# Photocatalytic Cleavage of β-*O*-4 Ether Bonds in Lignin over Ni/TiO_2_

**DOI:** 10.3390/molecules25092109

**Published:** 2020-04-30

**Authors:** Changzhou Chen, Peng Liu, Haihong Xia, Minghao Zhou, Jiaping Zhao, Brajendra K. Sharma, Jianchun Jiang

**Affiliations:** 1Key Lab. of Biomass Energy and Material, Jiangsu Province, National Engineering Lab. for Biomass Chemical Utilization; Key and Open Lab. on Forest Chemical Engineering, Institute of Chemical Industry of Forest Products, Chinese Academy of Forestry, SFA, Nanjing 210042, China; changzhou_chen@163.com (C.C.); liupengnl@163.com (P.L.); xiahaihong87@126.com (H.X.); zhaojiaping1017@163.com (J.Z.); jiangjc@icifp.cn (J.J.); 2Co-Innovation Center of Efficient Processing and Utilization of Forest Resources, Nanjing Forestry University, Nanjing 210037, China; 3Illinois Sustainable Technology Center, Prairie Research Institute, one Hazelwood Dr., Champaign, University of Illinois at Urbana-Champaign, Champaign, IL 61820, USA

**Keywords:** lignin, β-*O*-4, photocatalyst, Ni/TiO_2_, oxidant PCC

## Abstract

It is of great importance to explore the selective hydrogenolysis of β-*O*-4 linkages, which account for 45–60% of all linkages in native lignin, to produce valued-added chemicals and fuels from biomass employing UV light as catalyst. TiO_2_ exhibited satisfactory catalytic performances in various photochemical reactions, due to its versatile advantages involving high catalytic activity, low cost and non-toxicity. In this work, 20 wt.% Ni/TiO_2_ and oxidant PCC (Pyridinium chlorochromate) were employed to promote the cleavage of β-*O*-4 alcohol to obtain high value chemicals under UV irradiation at room temperature. The Ni/TiO_2_ photocatalyst can be magnetically recovered and efficiently reused in the following four consecutive recycling tests in the cleavage of β-*O*-4 ether bond in lignin. Mechanism studies suggested that the oxidation of β-*O*-4 alcohol to β-*O*-4 ketone by oxidant PCC first occurred during the reaction, and was followed by the photocatalysis of the obtained β-*O*-4 ketone to corresponding acetophenone and phenol derivates. Furthermore, the system was tested on a variety of lignin model substrates containing β-*O*-4 linkage for the generation of fragmentation products in good to excellent results.

## 1. Introduction

While fossil fuels were widely regarded as the primary source for chemicals and energy, the fraction of chemicals and fuels obtained from renewable resources, such as biomass, can be expected to be good alternatives in the future [1,2,3,4,5]. Lignin contains complex natural aromatic subunits (sinapyl, coniferyl, coumaryl), and diverse types of linkages (β-*O*-4, α-*O*-4 and 4-*O*-5) [6,7]. Among these types of ether bonds, β-*O*-4 is the most abundant linkage in lignin, resulting in a variety of studies focused on the cleavage of the β-*O*-4 bond employing lignin dimeric model compounds (Figure 1a) [8,9]. In the past decades, tremendous efforts have been devoted into the degradation of lignin by Hartwig [10], Baker [11], Dyson [12], Ellman [13], Barta [14], etc.

In general, lignin depolymerization processes always involve the utilization of either transition-metal catalysts [15] or noble-metal catalysts [16]. However, those approaches still remain significant challenges that have never been dealt with. Particularly, a typical lignin degradation process requires highly demanding reaction conditions (elevated temperature, in the presence of H_2_, etc.) [17,18]. Therefore, it has become increasingly important to develop an efficient catalytic system to address these issues [19]. Recently, the photocatalytic degradation of lignin—which has various advantages such as milder reaction conditions, simple reaction process without further purification, filtration, or solvent changes [20,21,22]—has received increasing attention, as it is regarded as a potential alternative to traditional process. For example, Stephenson et al. explored an efficient and innovative two-stage lignin degradation method, in which, [4-Acetamido-TEMPO]BF_4_ mediated the benzylic oxidation in the first step and followed by a photoredox-catalyzed reductive C-O cleavage utilizing [Ir(ppy)_2_(dtbbpy)]Pf_6_ as the photocatalyst [23]. Based on the previous study, a bimetal catalytic strategy was carried out in the practical and operationally simple two-step degradation of lignin from the same group involving Pd-catalyzed benzylic oxidation and photoredox-catalyzed ([Ir(ppy)_2_(dtbbpy)]PF_6_) reductive fragmentation for the efficient cleavage of the β-*O*-4 ketone obtained from the first step to generate lower-molecular-weight aromatic building blocks [24]. Besides [Ir(ppy)_2_(dtbbpy)]Pf_6_, a variety of photocatalysts showed an excellent ability in cleaving C-O bonds in lignin, including *fac*-Ir(ppy)_3_ [25], [Ir(ppy)_2_(dtbbpy)]PF_6_ [26], [Ir{dF(CF_3_)_2_ppy}_2_(dtbbpy)]PF_6_ [27]. However, these catalysts needed to be stored under anhydrous and anaerobic conditions, and could not be recovered for the next run. Semiconductor, such as TiO_2_, attracted the attention of many researchers due to its versatile advantages involving stable catalytic performance and low cost [28,29,30]. It has been widely reported that the introduction of transition metal [31,32,33] or noble metal nanoparticles [34,35,36] to TiO_2_ support is beneficial for electron transfer in the photochemical reaction process. For instance, Farnood et al. demonstrated a novel Bi and Pt co-modified TiO_2_ catalyst for the photo-oxidation of lignin under solar light for the generation of guaiacol, vanillic acid, vanillin and 4-phenyl-1-buten-4-ol [37]. Srisasiwimon et al. employed lignin-based carbon to modify TiO_2_ for the generation of a composite photocatalyst (TiO_2_/lignin), then supported Pt was prepared and exhibited excellent catalytic activity for the production of high value chemicals from lignin [38]. However, noble metal-based catalysts will inevitably add to the cost and always suffer from detaching issues; the development of non-noble metal based photocatalysts and the recovery of photocatalysts still meet huge challenges. Therefore, it is of great significance to develop a simple and efficient catalyst to address this issue.

The degradation of organics over TiO_2_-supported catalysts is essentially a free radical reaction process [39]. When irradiated with ultraviolet light (wavelength < 400nm and energy band = 3.2 eV), an electron in the valence band is excited and advances to the conduction band for the generation of a photo-generated electrons (e^−^), leaving photo-generated holes (h^+^) in the valence band (Figure 1b). The obtained photo-generated holes (h^+^) have an excellent ability to oxidize organic chemicals attached to the surface of TiO_2_ or oxidize OH^−^ to yield ^·^OH radical in the first place, and subsequently oxidize the organics for small molecule compounds [40,41,42]. In this work, TiO_2_-supported nickel catalyst was introduced into the selective oxidation of β-*O*-4 ketone model compounds, resulting in the generation of value-added aromatics. Afterwards, a two-step process was successfully achieved for the photocatalytic cleavage of β-*O*-4 alcohols. In the first step, β-*O*-4 alcohol was oxidized into β-*O*-4 ketone by PCC oxidant; then, the cleavage of the β-*O*-4 bond happened over Ni/TiO_2_. Through the screening of light sources, solvents and oxidations, herein we reported a super mild reaction condition (30W UV, iPrOH, and PCC) for the cleavage of the β-*O*-4 bond in lignin model compounds. Finally, the possible reaction mechanism was also proposed. The basic physicochemical properties were investigated by means of XRD, TEM, and XPS analyses.

## 2. Result and Discussion

### 2.1. Catalyst Characterization

XRD spectra of pure TiO_2_ and 20 wt.% Ni/TiO_2_ are presented in Figure 2i. The XRD pattern of pure TiO_2_ was clearly shown in the pattern (a). The characteristic peaks of anatase and rutile crystals of TiO_2_ could be obviously discovered, in which peaks of 2θ = 53.9° and 62.9° marked by Miller indices (210) and (002) belonged to the rutile crystal, and the other peaks of 2θ = 25.4°, 37.9° and 48.2° marked by Miller indices (101), (004) and (200) belonged to the anatase TiO_2_ [43]. Moreover, the peak at 27.5° was too weak to be observed which belonged to rutile crystal. In the pattern of 20 wt.% Ni/TiO_2_, all the characteristic peaks of anatase and rutile crystals of TiO_2_ could be clearly found, and three characteristic peaks at 2θ of 44.6°, 52.0° and 76.6°, marked by Miller indices (111), (200) and (220), could also be observed at relatively high angles, indicating the presence of metallic nickel. No NiO species were observed in the XRD pattern, which proved that metallic nickel and TiO_2_ could be stably stored without oxidation in air before the reaction process [44]. The dispersion and average particle size of the photocatalyst Ni/TiO_2_ were characterized by TEM, which was also presented in Figure 2 (Figure 2ii and iii). It could be clearly seen that small particles, representing clusters of metallic Ni, were homogenously dispersed on the surface of TiO_2_ particles, and the average size of Ni particles was 13.45 nm or so.

The surface element compositions of 20 wt.% Ni/TiO_2_ photocatalyst were analyzed by XPS. The survey scan and XPS patterns of Ti2p and Ni2p were presented in Figure 3. The general spectra (Figure 3iii) exhibited the presence of respective metals. In Figure 3i, the binding energies at 852.5 eV and 869.0 eV might ascribe to Ni^0^(2p_3/2_) and Ni^0^(2p_1/2_), respectively, and the binding energies at 855.0 eV and 874.5 eV belonged to the main line of Ni^2+^(2p_3/2_), and Ni^2+^(2p_1/2_), which indicated the presence of both metallic Ni and NiO on the surface of TiO_2_. It could be seen from Figure 3ii, that the peaks at 458.5 and 464.2 eV were ascribed to Ti^4+^(2p_3/2_) and Ti^4+^(2p_1/2_) in TiO_2_, respectively [43].

### 2.2. Optimization of the Reaction Condition

Above all, the amount of Ni/TiO_2_ could affect the efficiency of the cleavage of β-*O*-4 ether. 2-phenoxy-1-phenylethan-1-one (**1a**) was selected as a model compound, and the results were described in Table 1. As expected, the conversion increased with the increasing catalyst amount (from 0 to 30 wt.% Ni/TiO_2_), and the changes of acetophenone and phenol yields indicated the same trend. Meanwhile, 20 wt.% Ni/TiO_2_ could achieve a total conversion of 2-phenoxy-1-phenylethan-1-one in the photocatalytic system (Table 1, entry 1). Therefore, 20 wt.% Ni/TiO_2_ was employed in the following experiments. 

In addition, the influence of light source on the photo-catalysis of β-*O*-4 ketone was conducted. To evaluate whether the light source played a significant role in the cleavage of C-O bond in the lignin, comparative tests were carried out in different light sources (darkness, sunlight, 30 W UV, and 30 W blue LED). As expected, the transformation of **1a** failed in the darkness and no acetophenone and phenol were observed in GC/MS (Table 2, entry 1). Meanwhile, sunlight catalytic process achieved the same results and 100% recovery of **1a** was observed (Table 1, entry 2). The above phenomena in Table 2 (entry 1 and entry 2) were reasonable, as previous studies reported that the lignin degradation reaction generally needed to be performed under elevated temperature, high pressure and even catalyzed by homogeneous and/or heterogeneous catalysts. However, the transformation of **1a** at 180 °C under sunlight achieved a little improvement with a yield of 11% and 8% toward acetophenone and phenol, respectively (Table 2, entry 3). In comparison, it was amazing to find that the cleavage of β-*O*-4 C-O bond was remarkably expedited under UV irradiation, while the yield of acetophenone and phenol increased sharply to about 82% and 80%, respectively, indicating an excellent efficiency for the cleavage of β-*O*-4 ketone bond in lignin (Table 2, entry 4). Apart from darkness, sunlight and UV, blue LED was also selected as the light source in the 20 wt.% Ni/TiO_2_ catalytic system. Similar to sunlight, blue LED also seemed to not be suitable for the photocatalyzed cleavage of C-O bond in lignin (Table 2, entry 5). Due to the relatively long wavelength of blue LED (>400 nm), the electron in the valence band could not be excited and subsequently advanced to the conduction band for the generation of a photo-generated electrons (e^−^) to leave a photo-generated holes (h^+^) [45], thus the reaction gave a very poor β-*O*-4 ketone conversion rate (Table 2, entry 5). The comparison results between different light sources suggested that the wavelength was a key influencing factor in the photolysis of β-*O*-4 model catalyzed by TiO_2_ supported catalysts. Therefore, we found 20 wt.% Ni/TiO_2_ performed the better catalytic activity for the cleavage of β-*O*-4 ketone under UV irradiation (30 W).

With the 20 wt.% Ni/TiO_2_ photocatalytic system under 30 W UV irradiation in hand, the influence of solvents on the cleavage of β-*O*-4 models was explored subsequently. The results in Table 3 indicate that 20 wt.% Ni/TiO_2_ exhibited an outstanding catalytic activity for the C-O bond cleavage, but the conversion and yields varied in different polar and apolar solvents. In the apolar solvent system, the photocatalytic process displayed a low conversion rate and low product yields (Table 3, entry 1 and 2), probably due to the lower solubility in solvent. Hydrogenation-donor solvents such as *i*PrOH and methanol were believed to favor the cleavage of β-*O*-4 models in lignin, while a conversion rate of nearly 100% was achieved in **1a**, accompanied by remarkably high yields of acetophenone and phenol (Table 3, entry 3 and 4). When carried out in DMF, the reaction process also showed an excellent conversion rate (100%) and yields (82% of acetophenone and 80% of phenol). When the reaction was conducted in acetonitrile or acetone, improved conversion rate was observed, which were higher than 60% (Table 3, entry 6 and 7). However, the yields of acetophenone and phenol were quite low (all below 38%), indicating a poor selectivity. H_2_O was expected to be a green and environmentally-friendly solvent in a variety of chemical reactions. However, relatively lower conversion rate (15%) and yields were achieved (6% toward acetophenone and 5% toward phenol) owing to poor solubility, limiting the occurrence of the photocatalytic reaction in water (Table 3, entry 8). In general, the catalytic performance of the solvents followed the below order: *i*PrOH > methanol > DMF > acetone > acetonitrile > *n*-hexane > cyclohexane > H_2_O. All of this taken together, the results above reflected that *i*PrOH was the optimal solvents for the C-O bond cleavage over 20 wt.% Ni/TiO_2_. 

### 2.3. Scope of the Substrates

In order to continue the exploration of the 20 wt.% Ni/TiO_2_ photocatalytic activity, a variety of substrates including β-*O*-4 linkage were chosen as model compounds under the optimal conditions. It could be clearly seen in Scheme 1 that, the main products in the above reactions were acetophenone and phenol derivates, suggesting the occurrence of the selective cleavage of the C-O bond in those β-*O*-4 model compounds. Substrates without any substituted groups on the benzene ring **1a** could achieve a 100% conversion rate, to give 88% of acetophenone and 82% of phenol in iPrOH. Methoxy-substitutions of *O*-terminus aryl led to no apparent decrease in yields compared to **1a**. It seemed that steric effect had no obvious influence in our photocatalytic system. Hence, sing-ortho-substituted ketone **1b** and bis-ortho-substituted ketone **1c** afforded ketone products **2b** (78%), **2c** (80%) and phenol products **3b** (77%), **3c** (78%), respectively, after 12 h. In addition, coumaryl-based substrates (**1d**–**1f**) were subsequently explored in the Ni/TiO_2_-catalyzed photocatalytic system. It was gratifying to discover that methoxy-substituted ketones **1d**–**1f** were tolerated with no apparent decrease in yields, affording phenol **3d**, guaiacol **3e** and syringol **3f** in 80%, 77% and 72%, respectively. Moreover, different leaving groups such as α-acetoxy group instead of phenol compared with **1d** exhibited a significant role in our photocatalyzed system, delivering trace amounts of 1-(4-methoxyphenyl)ethan-1-one. The same results were achieved when **1h** was employed in the process and nearly 100% recovery of substrate **1h** was observed in the GC/MS. Meanwhile, **1i** was carried out in our reaction system and 32% yield of ketone derivate and 52% yield of phenol derivate were obtained. Finally, the Ni/TiO_2_-photocatalytic system was almost not responsible for the fragmentation of the corresponding β-*O*-4 alcohol (**1j**) under the optimal reaction condition and almost no target ketone and phenol products were observed. Therefore, this was again confirmed that previous reports could not achieve one-pot degradation of β-*O*-4 alcohol models, instead, the process always involved two steps: noble mental catalyzed oxidation process and photoredox-catalytic reductive fragmentation of the β-*O*-4 linkage. The dominant reason for this phenomenon was that the homolytic bond dissociation enthalpy of β-*O*-4 ether bond has been theoretically predicted to decrease from 62.2 to 55.9 kcal/mol after one oxidation process, leading to the fragmentation of β-*O*-4 ketone much easier than β-*O*-4 alcohol [46].

In the past few years, a two-step method for the fragmentation of β-*O*-4 alcohol to valued-added aromatics had been widely explored. Stephenson and his team firstly developed a room-temperature lignin degradation strategy with a chemoselective benzylic oxidation [4-Acetamido-TEMPO]BF_4_, followed by the reductive C-O bond cleavage over Ir-based catalyst [23]. In 2017, Stephenson et al. proposed an electrocatalytic oxidation method, coupled with a photocatalytic catalyst (Ir catalyst) catalyzed cleavage for β-*O*-4 bond in lignin [47]. Subsequently, Stephenson et al. developed a novel and operationally simple two-step lignin degradation method, involving Pd-catalyzed aerobic oxidation and visible-light photoredox-catalyzed reduction for the efficient cleavage of β-*O*-4 alcohol in 2019 [24]. Taking all of these into consideration, we assumed whether it could be achieved in one-pot (Scheme 2). According to the special ability of Ni/TiO_2_ for the transformation of β-*O*-4 ketone to corresponding acetophenone and phenol, we attempted the mechanical mixing of oxidant and Ni/TiO_2_ for the C-O bond cleavage, and found that it was hardly efficient for the cleavage of β-*O*-4 alcohols. Herein, three oxidants ([4-Acetamido-TEMPO]BF_4_, H_2_O_2_, PCC) were employed as oxidants in the reaction under optimal conditions. Unfortunately, it seemed that it was impossible to achieve the cleavage of C-O bond in β-*O*-4 alcohols by adding the oxidants (herein [4-Acetamido-TEMPO]BF_4_, H_2_O_2_ and PCC) and photocatalyst (herein 20 wt.% Ni/TiO_2_) in a one-pot process (Table 4, entry 1–3). This was probably due to the interference with the Ni/TiO_2_ photocatalyst. As is well-known, PCC is a perfect oxidant to transfer primary and secondary alcohols to aldehydes and ketones in DCM, and we question whether we could take PCC, **1k** and photocatalyst Ni/TiO_2_ in DCM in one-pot. Disappointingly, one case using PCC as an oxidant and DCM as a solvent gave a perfect conversion of **1k** (100%), but failed to yield acetophenone and phenol (Table 4, entry 4). When the experiment was carried out in two steps, β-*O*-4 alcohol was firstly oxidized by PCC, and the obtained β-*O*-4 ketone was transferred to yield acetophenone (66%) and phenol (61%) (Table 4, entry 5). We also suspect that solvents played a crucial role in this reaction. Hence, a mixture solvent (iPrOH: DCM=1:1) was tried under the optimal reaction condition and the same results were achieved—no acetophenone and phenol were detected. We suspected that the most likely reason was that the oxidant and reductant are mixed together, and that they could react with each other with no useful outcome. (Table 4, entry 6). Taken together, we continued to explore a two-step approach to achieve the hydrogenolysis of the β-*O*-4 alcohols of lignin-derived compounds (PCC oxidation followed by Ni/TiO_2_ photocatalysis).

Inspired by the excellent performance of two-step strategy, that is PCC oxidation in the first step followed by 20 wt.% Ni/TiO_2_ photocatalysis under the optimal reaction condition for the efficient cleavage of the β-*O*-4 bond. Then, the hydrogenolysis of C-O bond was further studied in more detail by varying the substituent groups on the benzene ring (Scheme 3). Owing to the successful two-step strategy for the transformation of β-*O*-4 alcohol to corresponding aromatics. However, the yields of aromatics exhibited only a slight decrease in Scheme 3, due to the loss of substrates in the two-step process. When no substituted groups on the benzene ring, **1k** could reach a high conversion to yield 66% of acetophenone and 61% of phenol. Methoxy substitutions of *O*-terminus aryl resulted in no apparent decrease in yields compared to **1k**, which was similar to the above study of β-*O*-4 ketones. Therefore, sing-ortho-substituted alcohol **1l** and bis-ortho-substituted alcohol **1m** delivered ketone products **2l** (62%), **2m** (64%) and phenol products **3l** (65%), **3m** (58%), respectively, after 12 h at room temperature. Moreover, coumaryl-based system (**1n**–**1p**) was soon carried out in the two-step Ni/TiO_2_-mediated photocatalytic strategy. It was satisfactory to find that methoxy-substituted alcohols **1n**–**1p** were tolerated with no apparent decrease in yields, delivering phenol **3n**, guaiacol **3o** and syringol **3p** in a yield of 66%, 58% and 49%, respectively. Based on the previous study of β-*O*-4 model compound (**1j**) in Scheme 2, we attempted the two-step strategy for the transformation of **1j**. Unfortunately, oxidation PPC showed an excellent activity to turn α-hydroxyl group and γ-hydroxyl group to corresponding ketone and aldehyde, resulting in the failed generation of **2q**. Above all, it could be verified that the two-step strategy was highly efficient for the transfer hydrogenolytic cleavage of the β-*O*-4 bond in a variety of lignin-derived model compounds.

It was of great importance to understand the pathway for the photocatalytic degradation of β-*O*-4 ether bond in lignin. Generally, the hydroxyl substituted on the α-C was firstly turned to ketone through the PCC oxidation process (①), and subsequently, the generated β-*O*-4 ketone underwent a C-O bond cleavage to deliver the desired ketone and phenol fragmentation products (②–④) after hydrogen atom abstraction (⑤) and protonation (⑥). This was in accordance with Enright’s study, which put forward the oxidation and photochemical reduction of a lignin model substrate from benzylic alcohol to guaiacol and 4-methoxyacetophenone via a benzylic ketone intermediate [21]. In addition, the reusability of Ni/TiO_2_, based on the recycling tests under the optimal reaction conditions, was later investigated by employing **1k** conversion as a model reaction. The photocatalyst Ni/TiO_2_ could maintain a good catalytic activity after four successive runs and almost no change in the yields of corresponding aromatics was observed. Apart from this, 20 wt.% Ni/TiO_2_ could be recovered magnetically in the end (Figure 4), which could facilitate the recycling process.

Although the trial for the one-step β-*O*-4 alcohol reduction was a failure, a two-step process was employed for the photocatalytic cleavage of β-*O*-4 alcohols: PCC oxidation followed by photoreduction. Mechanical mixing of oxidant and photocatalyst cannot achieve an ideal effect. Therefore, it is important to seek an efficient catalyst that can both oxidize alcohol and break the C-O bond.

## 3. Experimental

### 3.1. Materials

TiO_2_ were purchased from Aladdin Industrial Inc. Shanghai, China and used without further treatment. *N*,*N*-dimethylformamide was obtained from Alfa Aesar Reagent Co., Ltd. (Shanghai, China). Ni(NO_3_)_2_^.^6H_2_O was provided from Aladdin Industrial Inc. Shanghai, China. β-*O*-4 ketones (**1a–1f**) were synthesized according to the previous literature [48]. In a typical process of 2-phenoxy-1-phenylethan-1-one (**1a**), a 500 mL round bottom flask equipped with a reflux condenser and a dropping funnel was charged with phenol (5.2 g, 55 mmol) and potassium carbonate (10.4 g, 76 mmol) in acetone (250 mL) and stirred at room temperature. To this solution, 2-bromoacetophenone (10.0 g, 50 mmol) in acetone (50 mL) was added dropwise over 10 min at room temperature. The resulting suspension was stirred at reflux for 5 h. After the reaction, the suspension was filtered and concentrated in vacuo. The crude product was purified by recrystallization from petroleum ether to obtain 2-phenoxy-1-phenylethan-1-one as a white solid (10.5 g, 49 mmol) in a 98% yield. Moreover, β-*O*-4 ketones (**1k–1p**) was synthesized through a reductive method, in which NaBH_4_ was employed as the reductant. In addition, substrates **1g–1j** was obtained from leyan.com.cn. 

### 3.2. General Procedure for Ni/TiO_2_ Catalyzed β-O-4 Model Compounds

In a typical catalytic reaction, 100 mg of 2-phenoxy-1-phenylethan-1-one or other derivates, 20 mg of Ni/TiO_2_ catalyst, and 2.5 mL *N*,*N*-dimethylformamide were placed into a glass tube (10 mL). The photocatalytic reaction was performed under 30 W ultraviolet light (main wavelength at around 395 nm) in air at room temperature. At the end of the reaction, the mixture solution was filtered to collect the catalyst and the filtrate was analyzed by the Gas Chromatograph/Mass Spectrometer (GC/MS, Agilent 7890, Shanghai, China) using *n*-dodecane as an internal standard. The collected catalyst was washed with water and ethanol for three times and dried at 105 °C for the next run under the optimal reaction condition. The conversion and product yields in the liquid phase were calculated according to the following formula, respectively:(1)$$Conversion=\frac{mole\;of\;reacted\;substrate}{total\;mole\;of\;substrate\;feed}*100\%$$
(2)$$Yield\;of\;acetophenone=\frac{mole\;of\;acetophenone}{total\;mole\;of\;substrate\;feed}*100\%$$
(3)$$Yield\;of\;phenol=\frac{\;mole\;of\;phenol}{total\;mole\;of\;substrate\;feed}*100\%$$

### 3.3. Catalyst Preparation

Ni/TiO_2_ was prepared by the traditional impregnation method. In a typical process of 20 wt.% Ni/TiO_2_, Ni(NO_3_)_2_^.^6H_2_O (0.2 g) and TiO_2_ (2.0 g) were added in deionized water and stirred for 24 h. Then, the suspension was dried for 12 h at 105 °C in the oven. Subsequently, obtained grey solid was calcined at 500 °C for 2 h in the muffle furnace, and then reduced in a tube furnace under hydrogen atmosphere at 500 °C for 2 h.

### 3.4. Catalyst Characterization

Powder X-ray diffraction (XRD) was conducted on a Bruker D8 Advance X-ray powder diffractometer (Shanghai, China). Transmission electron microscopy (TEM) images were collected using a TEM Tecnai G2 20 (Thermo-VG Scientific, Shanghai, China). The X-Ray photoelectron spectroscopy (XPS) was examined on an ESCALAB-250 (Thermo-VG Scientific, Shanghai, China) spectrometer with Al Kα (1486.6 eV) irradiation source.

## 4. Conclusions

In summary, we have shown a mild and efficient two-step strategy that proceeded through the oxidation of β-*O*-4 alcohol to ketone first, then 20 wt.% Ni/TiO_2_-photocatalysis for the transformation of β-*O*-4 ketone to obtain value-added chemicals. The Ni/TiO_2_ photocatalyst not only had an excellent ability to catalyze the transfer hydrogenolytic cleavage of β-*O*-4 ether bonds in diverse substrates under UV irradiation (30 W), but could also be easily recovered magnetically from the reaction process for the next four recycling tests. Our photocatalytic system was also suitable for the cleavage of various β-*O*-4 alcohols in lignin, and the basic physicochemical characterization illustrated that the high activity of photocatalyst originated from the metallic Ni on the surface of TiO_2_. It is worth noting that even though the two-step strategy for the cleavage of β-*O*-4 bond in lignin performed well, it still remains a great challenge to realize the hydrogenolysis in one-pot. This work may inspire more researches on the depolymerization of lignin using photocatalysts in one-pot.

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
