# Peer review of "Photocatalytic Cleavage of β-O-4 Ether Bonds in Lignin over Ni/TiO2"

_molecules, 2020, doi:10.3390/molecules25092109_

Round 1

Reviewer 1 Report

In this work, Ni/TiO2 was first introduced into the selective oxidation of β-O-4 ketone model compounds to form value-added aromatics. Then, a two-step process was for the photocatalytic cleavage of β-O-4 alcohols: PCC oxidation followed by photoreduction. Although the failure on the trial for the one-step β-O-4 alcohol reduction, this work had demonstrated the knowledge of the photoconversion with hydrogenolysis. It can be accepted after minor corrections on English writing and the question below.

I think with this work experience they can add one paragraph with comments about the one-pot problem.

Author Response

Response to Reviewer 1 Comments

Point 1: I think with this work experience they can add one paragraph with comments about the one-pot problem.

Response 1: First of all, thank you for your recognition of our manuscript, and we agree with your valuable opinions. So, we add one paragraph with comments about the one-pot problem.

“Although the failure on the trial for the one-step β-O-4 alcohol reduction, a two-step process was employed for the photocatalytic cleavage of β-O-4 alcohols: PCC oxidation followed by photoreduction. Mechanical mixing of oxidant and photocatalyst cannot achieve an ideal effect. Therefore, it is important to seek an efficient catalyst that can both oxidize alcohol and break the C-O bond.”

Reviewer 2 Report

The article entitled "Photocatalytic Cleavage of β-O-4 Ether Bonds in Lignin over Ni/TiO” by Changzhou Chen et al. deals with an interesting topic since selective cleavage of β-O-4 linkages is a crucial step for the effective valorization of lignocellulosic biomass. However, I think that the manuscript should be improved in some aspects before publication on Molecules as reported in the following:

  • The authors claim several times in the manuscript that they propose a green and environmentally-friendly approach, however I do not agree with this statement. Indeed, in the first step of their process (conversion to ketone) they employ PCC (Pyridinium chlorochromate), which is harmful for the human health and the environment and, then, the photocatalytic reaction is performed in toxic solvents, while in water the yields and conversion rates are limited. Other authors proposed more environmentally-friendly approaches exploiting also visible light instead of UV light (see e.g. Nat. Catal. 1, 772-780 (2018)).
  • Commenting the XRD patterns in Figure 2, the authors says (page 4, line 133): “the peaks at 2θ=9° and  62.9°  marked  by  Miller  indices (210) and (002) belonged to the rutile crystal”. However, the most intense XRD reflection for rutile (the 110, see e.g. J. Phys. Chem. C 2013, 117, 11186 − 11196) is expected to occur at 27.5°: why is it not observed and commented? Moreover, in the caption of Fig. 2 “XRD spectra” should be substituted with “XRD patterns” since XRD is not a spectroscopy.
  • The English of the manuscript should be improved and the typos present corrected. For the sake of brevity, I report only one emblematic example: (i) page 2, line 64, “or noble MENTAL nanoparticles” should be substituted with “or noble METAL nanoparticles”.

Author Response

Response to Reviewer 2 Comments

Point 1: The authors claim several times in the manuscript that they propose a green and environmentally-friendly approach, however I do not agree with this statement. Indeed, in the first step of their process (conversion to ketone) they employ PCC (Pyridinium chlorochromate), which is harmful for the human health and the environment and, then, the photocatalytic reaction is performed in toxic solvents, while in water the yields and conversion rates are limited. Other authors proposed more environmentally-friendly approaches exploiting also visible light instead of UV light (see e.g. Nat. Catal. 1, 772-780 (2018)).

Response 1: In our work, Ni/TiO2 was introduced into the selective oxidation of β-O-4 ketone model compounds to form value-added aromatics. Then, a two-step process was for the photocatalytic cleavage of β-O-4 alcohols: PCC oxidation followed by photoreduction. Although the failure on the trial for the one-step β-O-4 alcohol reduction, this work had demonstrated the knowledge of the photoconversion with hydrogenolysis.

In the manuscript, statements about " green and environmentally-friendly approach " were not reasonable enough. Actually, it was a two-step process for the photocatalytic cleavage of β-O-4 alcohols under UV light. So, we modified the statement about " green and environmentally-friendly approach " in the full manuscript.

In addition, work on more environmentally-friendly approaches on the cleavage of β-O-4 alcohols will be soon carried out subsequently. We here thank you for the literature you provided, and we have already read this article carefully. This has provided tremendous help for us to carry out the next experiments in the future, and we once again express our deep gratitude to you.

Point 2: Commenting the XRD patterns in Figure 2, the authors says (page 4, line 133): “the peaks at 2θ=9° and 62.9°  marked  by  Miller  indices (210) and (002) belonged to the rutile crystal”. However, the most intense XRD reflection for rutile (the 110, see e.g. J. Phys. Chem. C 2013, 117, 11186 − 11196) is expected to occur at 27.5°: why is it not observed and commented? Moreover, in the caption of Fig. 2 “XRD spectra” should be substituted with “XRD patterns” since XRD is not a spectroscopy.

Response 2: We have carefully read the article you provided (J. Phys. Chem. C 2013, 117, 11186 − 11196), and it is sure that the most intense XRD reflection for rutile is expected to occur at 27.5°. However, the intensity of the peak at 27.5° is too weak to be observed. Due to the impact of COVID-19, we have no enough time to retest the XRD of this sample within 5 days. So we made a supplementary explanation for this problem in the manuscript. 

In addition, “XRD spectra” was be substituted with “XRD patterns”.

Point 3: The English of the manuscript should be improved and the typos present corrected. For the sake of brevity, I report only one emblematic example: (i) page 2, line 64, “or noble MENTAL nanoparticles” should be substituted with “or noble METAL nanoparticles”.

Response 3: We appreciate your valuable comments, and at the same time we apologize for our carelessness in preparation of this manuscript. Here we check the full manuscript, and improve the English of the manuscript. In addition to the typos in the full text, we also made changes to the inappropriate expressions in the full manuscript.

Reviewer 3 Report

The authors have attempted a conversion of an important biomass lignin into value added aromatic products. They have presented a good account of past work and outlined the past efforts of researchers to address the same problem in a very concise and understandable manner. However, here are some questions that i recomment the authors to address in the manuscript.

  1. PCC is a well known oxidant. However environmental chromium is considered to be detrimental. Can there be a possible e justification of its use here. I recommend the authors to kindly refer the recent work on electrochemical oxidation of alcohols (secondary)- https://www.nature.com/articles/s41467-019-10928-0 -Nature Communications volume 10, Article number: 2796 (2019) - I still believe that Ni/TiO2 is the primary focus of this manuscript. 
  2.  Can the authors justify the use of Ni/TiO2 and PCC in single pot. Because it is known that when oxidant and reductant are mixed together, they can react with each other with no useful outcome. - I recommend to change the argument that Ni/TiO2 was wrapped in mass of PCC. 
  3. Can the authors determine the VB and conduction band or the band gap of Ni/TiO2 using Cyclic voltametry? ( Chem. Commun., 2013, 49, 10742-10744 Supporting info) - This could justify the use of UV as excitation source for hydrogenolysis.
  4. In Table 4, the authors should state the type of reaction (if its a one pot or two step) in a dedicated additional column -  the results would look redundent to general audience. 

Author Response

Response to Reviewer 3 Comments

Point 1: PCC is a well known oxidant. However environmental chromium is considered to be detrimental. Can there be a possible e justification of its use here. I recommend the authors to kindly refer the recent work on electrochemical oxidation of alcohols(secondary)- https://www.nature.com/articles/s41467-019-10928-0 -Nature Communications volume 10, Article number: 2796 (2019) - I still believe that Ni/TiO2 is the primary focus of this manuscript.

Response 1: In our work, Ni/TiO2 was introduced into the selective oxidation of β-O-4 ketone model compounds to form value-added aromatics. Then, a two-step process was for the photocatalytic cleavage of β-O-4 alcohols: PCC oxidation followed by photoreduction. Although the failure on the trial for the one-step β-O-4 alcohol reduction, this work had demonstrated the knowledge of the photoconversion with hydrogenolysis.

However, work on electrochemical oxidation of β-O-4 alcohols will be soon carried out subsequently. We here thank you for the literature you provided, and we have already read this article carefully. This has provided tremendous help for us to carry out the next experiments in the future, and we once again express our deep gratitude to you

Point 2: Can the authors justify the use of Ni/TiO2 and PCC in single pot. Because it is known that when oxidant and reductant are mixed together, they can react with each other with no useful outcome. - I recommend to change the argument that Ni/TiO2 was wrapped in mass of PCC.

Response 2: We think what you said makes sense. The oxidant and reductant can react with each other. we follow your advice and change the argument that Ni/TiO2 was wrapped in mass of PCC in the manuscript.

Point 3: Can the authors determine the VB and conduction band or the band gap of Ni/TiO2 using Cyclic voltametry? ( Chem. Commun., 2013, 49, 10742-10744 Supporting info) - This could justify the use of UV as excitation source for hydrogenolysis.

Response 3: We are very sorry that we do not have enough time to determine the VB and conduction band or the band gap of Ni/TiO2 using Cyclic voltammetry within 5 days, because of the impact of COVID-19 that the Lab was closed. However, we have carefully read the article you provided (Chem. Commun., 2013, 49, 10742-10744 Supporting info). We have learned a lot in this article, which has provided great help for our future photocatalytic work.

Point 4: In Table 4, the authors should state the type of reaction (if its a one pot or two step) in a dedicated additional column -  the results would look redundent to general audience.

Response 4: We appreciate your valuable comments, and at the same time we apologize for the carelessness in the description of Table 4. So the caption of Table 4 was supplemented in detail.

Table 4. Optimization of the oxidants for the cleavage of β-O-4 alcohols in one-pot.a